# Fungal networks shape dynamics of bacterial dispersal and community assembly in cheese rind microbiomes

Yuanchen Zhang[1], Erik K. Kastman[1], Jeffrey S. Guasto[2] & Benjamin E. Wolfe[1]

Most studies of bacterial motility have examined small-scale (micrometer–centimeter) cell dispersal in monocultures. However, bacteria live in multispecies communities, where interactions with other microbes may inhibit or facilitate dispersal. Here, we demonstrate that motile bacteria in cheese rind microbiomes use physical networks created by filamentous fungi for dispersal, and that these interactions can shape microbial community structure. *Serratia proteamaculans* and other motile cheese rind bacteria disperse on fungal networks by swimming in the liquid layers formed on fungal hyphae. RNA-sequencing, transposon mutagenesis, and comparative genomics identify potential genetic mechanisms, including flagella-mediated motility, that control bacterial dispersal on hyphae. By manipulating fungal networks in experimental communities, we demonstrate that fungal-mediated bacterial dispersal can shift cheese rind microbiome composition by promoting the growth of motile over non-motile community members. Our single-cell to whole-community systems approach highlights the interactive dynamics of bacterial motility in multispecies microbiomes.

---

[1] Department of Biology, Tufts University, 200 Boston Avenue, Medford, MA 02155, USA. [2] Department of Mechanical Engineering, Tufts University, 200 College Avenue, Medford, MA 02155, USA. Correspondence and requests for materials should be addressed to B.E.W. (email: benjamin.wolfe@tufts.edu)

Multispecies microbial communities (microbiomes) play key roles in agricultural productivity, human health, and ecosystem services[1–3], but our understanding of the ecological processes and mechanisms that structure the diversity of microbiomes is still in its infancy[4–6]. Small-scale (micrometer–centimeter) dispersal of bacterial cells is one key ecological process that may impact the dynamics of microbial community assembly. After a propagule (cell, spore, etc.) of a microbial species colonizes a potential habitat, the ability to grow and rapidly spread may determine both the distribution and functions of that particular species within the community.

Many bacteria use active motility, via extracellular appendages or secreted metabolites, to disperse over small spatial scales up or down gradients of resources or attractants[7,8]. A significant body of work from just a few model bacterial species has determined the genetic and biophysical mechanisms of active bacterial dispersal, including swimming, swarming, gliding, twitching, and sliding[8,9]. Almost all of these studies have used monocultures of bacteria in highly simplified laboratory environments to dissect modes and mechanisms of bacterial motility. How these bacterial motility mechanisms, discovered in highly idealized laboratory systems, translate to complex multispecies microbiomes where microbes interact is largely unknown.

Changes in the abiotic and biotic environment, due to interactions with neighboring microbial species, have the potential to alter modes and mechanisms of bacterial motility and subsequent dispersal dynamics. Metabolites secreted into the environment by neighboring species may act as chemoattractants that can direct cell movement[10,11] or alter quorum sensing[10]. Modification of the physical environment by neighboring microbes could also impact cell dispersal. Solid surfaces can exert forces on swimming cells and guide them over long distances[12,13]. These same forces facilitate interactions between cells in biofilms, which can result in collective cell motility and dispersal[14,15].

One potentially widespread interaction that may shape the dynamics of bacterial cell dispersal is the migration of bacterial cells on fungal hyphae[16,17]. Multicellular filamentous fungi form mycelial networks that enable bacteria to migrate across simplified environments[16,18]. The specific biological and physical mechanisms underlying these interactions are not fully understood, but fungi likely maintain microenvironments that allow motile bacteria to swim and/or swarm in otherwise dry conditions[19]. Most previous studies characterizing fungal-mediated bacterial dispersal relied on artificial combinations of bacteria and fungi with unknown natural histories and limited ecological contexts[16,20,21]. The taxonomic breadth of bacteria that can disperse on fungal networks is also poorly characterized because prior work has largely focused on a limited number of bacteria and fungi in soil systems[16,19,21–23]. Moreover, the potential contribution of these strong, pairwise bacterial–fungal interactions to the assembly of microbiomes has not been tested. Fungal networks may shape the composition of bacterial communities by promoting the dispersal and growth of motile bacteria over non-motile community members.

Cheese rind biofilms are an ideal system for exploring the mechanisms and consequences of fungal-mediated bacterial dispersal in multispecies microbiomes. Rind biofilms form on the surfaces of cheeses that are aged in caves around the world, and several different genera of filamentous fungi commonly co-occur with motile Proteobacteria in cheese rinds[24,25,26,27]. Ecological dynamics in cheese rinds are easy to dissect due to the limited diversity of these microbiomes and the ability to culture most bacterial and fungal species that grow in these communities[27]. Previous work has demonstrated that strong bacterial–fungal interactions occur in cheese rinds[27–29], but mechanisms underlying these interactions are largely unknown.

Here we report the patterns, mechanisms, and consequences of bacterial dispersal on fungal networks in cheese rind microbiomes. We focus on one common cheese rind bacterium, *Serratia proteamaculans*, to characterize the mechanisms of bacterial dispersal on different fungal networks. We then place these pairwise interactions in an ecological context and quantify how fungal networks can shape the composition of multispecies cheese rind communities through dispersal facilitation. Our work highlights the ability of diverse cheese Proteobacteria to disperse on fungal networks and how fungal-mediated bacterial dispersal can promote the growth of motile bacteria over non-motile community members.

## Results

**S. proteamaculans disperses on cheese rind fungal networks.** During a culture-based survey of cheese rinds, we observed unusual streams of bacterial cells of the bacterium *S. proteamaculans* (strain BW106; hereafter *Serratia*) on hyphae of the filamentous fungus *Mucor lanceolatus* (strain SN1; hereafter *Mucor*) (Fig. 1, Supplementary Movie 1). These growth patterns suggested that *Serratia* used *Mucor* networks to disperse, possibly through the use of active motility mechanisms. To experimentally characterize this interaction, we first quantified bacterial dispersal using a co-spotting assay on standard lab media (brain heart infusion agar or BHI agar) with three different fungal networks: *Mucor*, *Galactomyces geotrichum* (hereafter *Galactomyces*), and a *Penicillium* strain closely related to *P. commune* (hereafter *Penicillium*) (Supplementary Table 1). All three fungi were isolated from cheese rinds. We chose these three fungi because they are the dominant fungi in natural and washed rind cheeses[27,30], and they represent three different types of fungal networks: *Mucor* is a fast-growing fungus with diffuse network growth[31,32], *Galactomyces* is also a fast-growing fungus but forms a dense

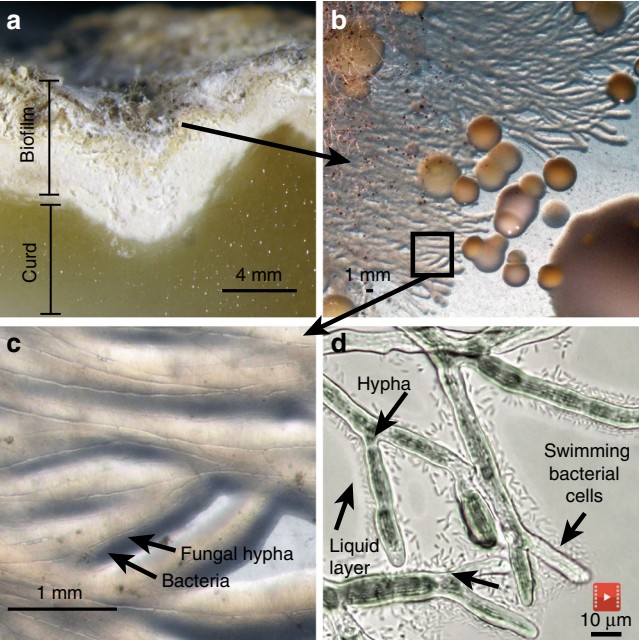

**Fig. 1** Bacterial dispersal on fungal networks in a cheese rind. **a** Cross-section of the French cheese Saint-Nectaire, showing the curd (paste) and the biofilm (rind). **b** Unusual streams of bacteria were observed when plating out this rind on plate count agar with milk and salt (PCAMS). **c** Closer examination revealed the bacterium *Serratia proteamaculans* growing along the hyphae of the fungus *Mucor lanceolatus*. **d** Individual swimming *Serratia* cells in the liquid layer surrounding *Mucor* hyphae (still image from Supplementary Movie 1)

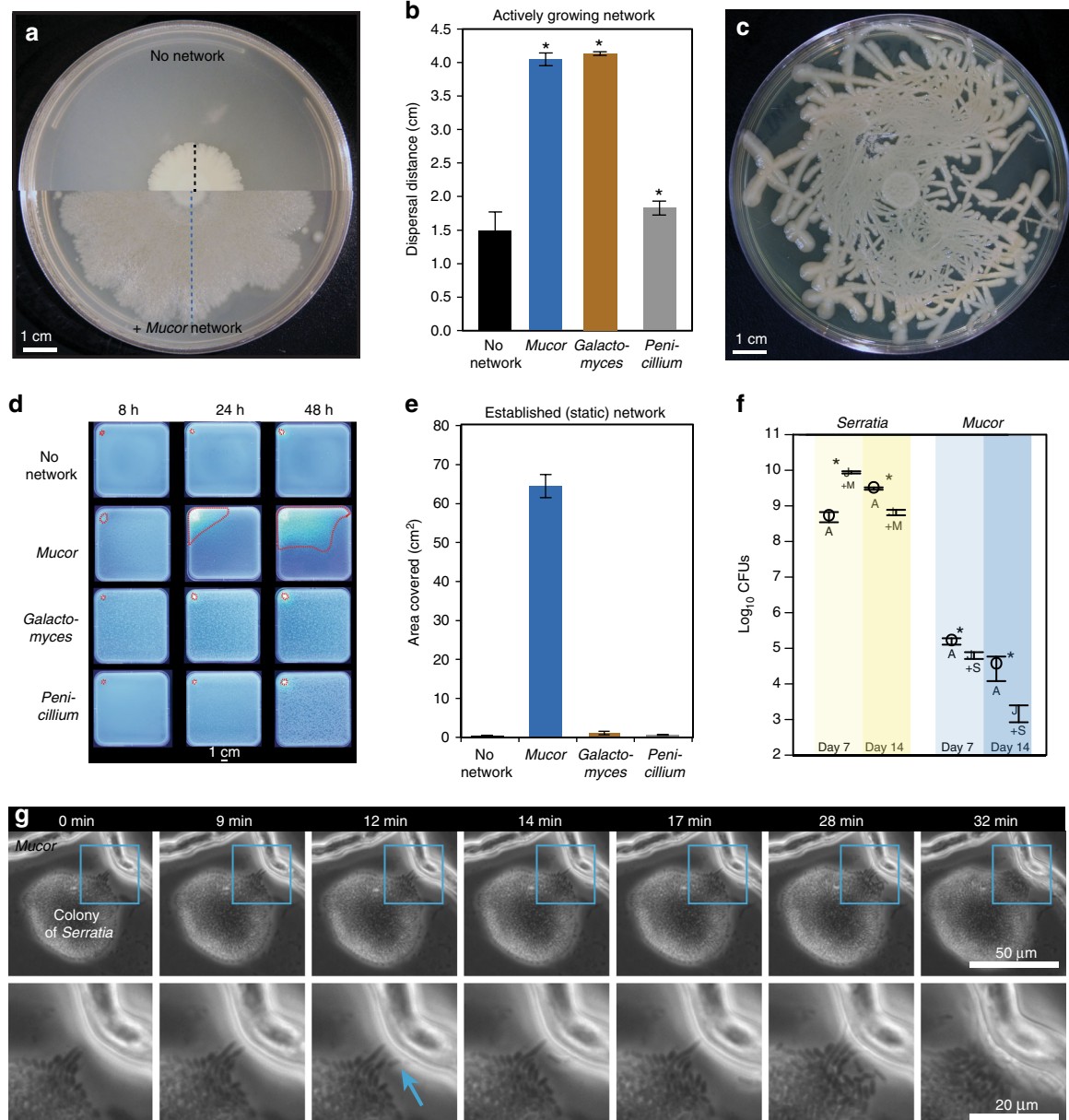

**Fig. 2** Macroscopic and microscopic characterization of *Serratia proteamaculans* dispersal on fungal networks. **a** Dispersal of *Serratia* on PCAMS agar without *Mucor* (top) and with *Mucor* (bottom). The black dashed line indicates dispersal distance alone and the blue dashed line indicates dispersal distance with a *Mucor* network (as displayed in Fig. 2b). **b** *Serratia* dispersal distance after 14 days of growth. Bars show mean distance from the center of the spot to the bacterial colony edge (±1 standard deviation, $n = 5$). Asterisks indicate significant differences in dispersal compared to control (Dunnett's test, $p < 0.05$). **c** Dispersal of *Serratia* along a synthetic fungal network made of glass fibers placed on PCAMS agar. **d** Representative time-lapse images after 8, 24, and 48 hours of GFP-tagged *Serratia* dispersing with no network or on an already established (static) network of *Mucor*, *Galactomyces*, or *Penicillium* on PCAMS agar. The outermost edges of the area where *Serratia* spread on each fungal network are outlined in red dashed lines. Each square Petri dish is 12 cm × 12 cm. **e** *Serratia* dispersal distance with no network or on an established (static) network of *Mucor*, *Galactomyces*, or *Penicillium*. Bars show the mean area covered (±1 s.d., $n = 3$). Area traveled on *Mucor* was significantly higher than the other fungal networks based on an ANOVA with a Tukey's post hoc test ($F_{3,15} = 1788$, $p < 0.0001$). **f** *Serratia* and *Mucor* growth, as measured by total colony forming units (CFUs), alone and in combination, after 7 and 14 days ($n = 5$). Asterisks indicate significant differences between Alone (A) and +*Mucor*/*Serratia* (+M/+S) (Student's *t*-test, $p < 0.05$). **g** Time lapse of *Serratia*–*Mucor* interaction over 32 minutes. The full colony of *Serratia* is shown in the top row, with two *Mucor* hyphae at the top. The bottom row is a zoomed-in section (outlined in blue) showing a close-up of pioneer *Serratia* cells colonizing the *Mucor* hyphae, indicated by the blue arrow. See Supplementary Figure 4 for more examples

network[32,33], and *Penicillium* is slow-growing and forms very dense fungal networks[32,34]. The cells of *Serratia* were co-spotted on BHI agar (1.5% agar) with each of these fungi or without a fungus ("No network", Fig. 2a). Fungal networks grew out from the co-spot, and *Serratia* was able to spread on the networks. After 14 days of incubation, the horizontal dispersal distance of

the bacterial colony from co-spot center to colony edge was quantified using a bacterial transfer approach (see Methods).

*Serratia* rapidly spread on networks of both *Mucor* and *Galactomyces*, with a 173% and 179% increase, respectively, in dispersal distance across the agar surface compared to *Serratia* without a fungal network (Fig. 2b). In contrast, *Penicillium*

networks provided limited dispersal facilitation of *Serratia*, with only a 23% increase in dispersal. The strong dispersal facilitation of *Serratia* was not limited to the environment on BHI agar; *Serratia* spread on *Mucor* networks on a range of media types, including cheese curd agar (CCA) (Supplementary Fig. 1). To confirm that this dispersal trait was not unique to our cheese strain, we quantified the ability of closely related *Serratia* strains and species from other environments (Supplementary Table 1) to spread on networks of *Mucor*. Most *Serratia* isolates showed substantial dispersal facilitation on *Mucor* networks, ranging from 145 to 175% increases in dispersal distance (Supplementary Fig. 2). Limited dispersal in a few isolates (e.g., *S. proteamaculans* strain B-41156, with only a 41% increase in dispersal distance) indicates natural variation in the ability of *Serratia* species to disperse on fungal networks.

Hyphae of filamentous fungi, including *Mucor* species, grow from the tip[35], and bacterial dispersal on fungal hyphae could be a result of passive dispersal when bacteria are pushed horizontally across surfaces by the growing fungi. To determine if *Serratia* spreads using active motility or passive dispersal by the fungus, a synthetic glass fiber network was created on top of a 10-μL spot of *Serratia* cells on BHI agar. These glass fibers are comparable in diameter (8 μm) to the hyphae of *Mucor* (10–25 μm) and provided a similar physical network for the movement of motile *Serratia* cells. After a week of growth, *Serratia* spread out from the initial spot and followed the topology of the synthetic network (Fig. 2c), suggesting that active motility drives the movement of this bacterium across physical networks.

In nature, bacterial cells may initially land in a microenvironment where an existing fungal mycelium is available for colonization. To determine if *Serratia* could disperse across established (static) fungal networks, we transformed *S. proteamaculans* BW106 with a green fluorescent protein (GFP)-producing plasmid to allow for real-time, non-destructive tracking of its spread across *Mucor*, *Galactomyces*, and *Penicillium* networks. Unlike the assays above that just track linear dispersal distance, this approach allowed us to track the network area covered by *Serratia*. As with the co-spot assays above, *Serratia* rapidly spread across existing networks of *Mucor*, covering an average of 64 cm$^2$ within 48 h, compared to 1 and 0.6 cm$^2$ on *Galactomyces* and *Penicillium* networks, respectively (Fig. 2d, e). The limited dispersal on static networks of *Galactomyces* contrasts with the high dispersal facilitation observed on actively growing *Galactomyces* networks (Fig. 2b). This discrepancy suggests that passive dispersal resulting from bacterial cells being pushed or dragged during fungal growth may contribute to dispersal on *Galactomyces* networks while dispersal on *Mucor* networks is largely due to active motility processes.

The substantial increase in dispersal distance of *Serratia* on *Mucor* networks may provide a significant benefit by allowing it to colonize unoccupied niches. But, changes in dispersal distance across a surface may not completely reflect the total impact on *Serratia* growth, as increased dispersal may actually decrease cell density. Moreover, bacterial dispersal data do not capture impacts on the fungal host. To measure the growth of both interacting partners, we determined the total colony-forming units (CFUs) of both *Serratia* and *Mucor* at 7 and 14 days of growth alone and in co-culture. As predicted from dispersal experiments, *Mucor* networks have a strong positive effect on *Serratia* growth at day 7, but this effect diminishes and results in growth inhibition at day 14 (Fig. 2f). The dispersal of *Serratia* on *Mucor* networks negatively impacts fungal growth at both day 7 and day 14. Surprisingly, unlike most other examples where *Serratia* species almost entirely eradicate the fungi with which they interact[36–38], *Mucor* is only partially inhibited by the bacterium and not completely killed (Supplementary Fig. 3).

Studies of bacterial dispersal on fungi have largely focused on macroscopic patterns (millimeter–centimeter scale) of bacterial and fungal hyphae growth[16,21,36]. But the initial phases of colonization occur at the micron scale and the relevant biophysical interactions that regulate bacterial dispersal on hyphae at these scales are poorly characterized. To identify how *Serratia* cells interact with *Mucor* networks at a microscopic scale, we used time-lapse microscopy of the two microbes co-cultured on a thin layer of BHI agar (≈0.7 mm in height). This approach revealed that interactions are initiated between individual cells of *Serratia* and a liquid layer surrounding *Mucor* hyphae (Fig. 2g). After imaging numerous bacterial–fungal contacts, we were able to consistently observe three phases of interaction initiation. First, the bacterial colony comes into physical contact with the liquid layer that surrounds the fungal hyphae as the hyphae grow near bacterial colonies. Next, pioneer cells at the edge of the bacterial colony rapidly transition from a stationary state to a motile state and swim in the liquid layer around the fungal hyphae. These pioneer cells swim along the fungal hyphae until they encounter physical barriers or reach an intersection and move to other hyphae. In the final phase, a mass of swimming cells colonizes the fungal hyphae. These interaction phases were commonly observed across replicate *Serratia*–*Mucor* contacts (Supplementary Fig. 4; Supplementary Movies 2 and 3).

The rapid cellular switch from a static colony state to a motile swimming state demonstrates that *Serratia* cells can quickly change behavior in the presence of the liquid layer around fungal networks. Collectively, these experiments and observations characterizing *Serratia*-fungal interactions from cheese rinds demonstrate that *Serratia* species can rapidly disperse on fungal networks via active bacterial motility mechanisms. We next sought to identify the mechanisms controlling these interactions and their ecological consequences for the assembly of these communities.

**Mechanisms driving *Serratia* dispersal on *Mucor* networks**. To determine potential genetic mechanisms that drive fungal-mediated dispersal facilitation of bacterial cells, we used three complementary approaches: (1) transcriptome sequencing (RNA-seq) to identify genes that are differentially expressed across the *Serratia* genome when grown on fungal networks, (2) transposon mutagenesis to identify genes that are essential for the dispersal phenotype described above, and (3) comparative genomics of different *Serratia* strains with variable dispersal abilities on *Mucor* networks. Given the rapid change from a stationary to motile cell population observed above, we predicted that the genes that control quorum sensing, flagellar biosynthesis, and other motility-related processes would be differentially expressed when *Serratia* was co-cultured with *Mucor* and would be essential for these interactions.

The presence of the fungal networks caused a shift in global gene expression of *Serratia* (Fig. 3a; Supplementary Data 1), with 108 genes showing significantly decreased expression and 41 genes with increased expression levels when *Serratia* was grown with *Mucor* networks. Surprisingly, the most differentially expressed genes were related to metabolic processes and other functions, not motility or quorum sensing (Fig. 3b). Of the 67 genes with decreased expression that had predicted functional annotations, almost half (33 genes) were predicted phage proteins. Significantly lower expression of genes associated with carbohydrate metabolism (8), amino acids and derivatives (4), and membrane transport (8), as well as genes associated with folate and biotin metabolism (4), suggests that growth on *Mucor* networks alters the supply of nutrients and vitamins available for *Serratia*. Surprisingly, we also detected downregulation of many

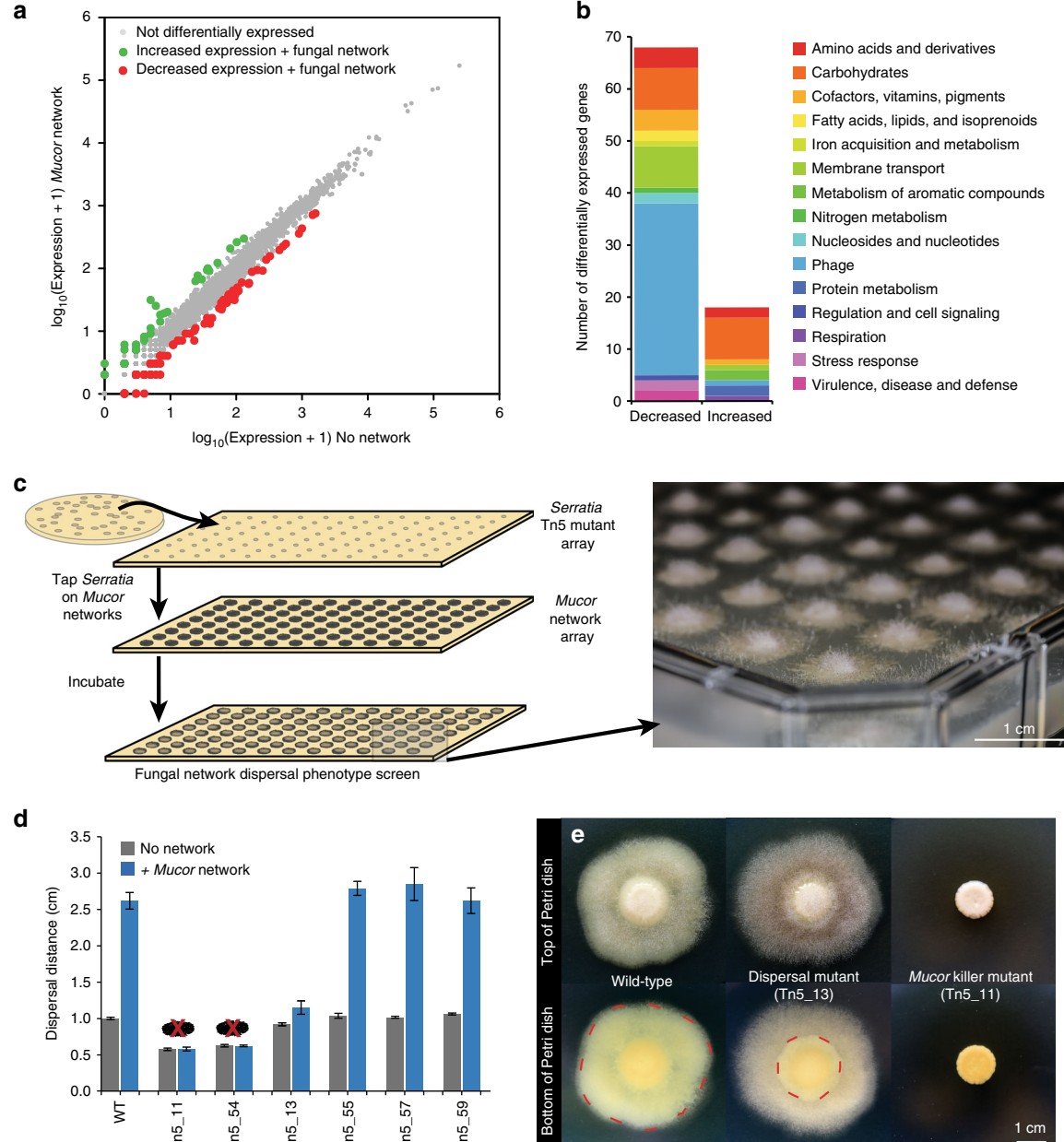

**Fig. 3** Genetic mechanisms driving *Serratia proteamaculans* dispersal on *Mucor* networks. **a** Biplot of mean gene expression data from an RNA-seq experiment of *Serratia* with and without *Mucor* networks. Genes that had greater than 2-fold change in expression and deemed significantly different based on a *q*-value of <0.05 (*p*-values adjusted for false discovery rate using the Benjamini–Hochberg procedure) were considered significantly increased or decreased in expression. Genes increased in expression are indicated in green, decreased in red, and no change in gray. Three biological replicates were performed for each treatment. **b** Functional distribution of differentially expressed genes that were annotated with a putative function. Functions were determined based on SEED annotations from RAST. Decreased = significantly decreased expression on *Mucor* networks. Increased = significantly increased expression on *Mucor* networks. **c** Illustration of the Tn5 transposon mutagenesis screen for *Serratia* dispersal on *Mucor*. *Serratia* mutants were arrayed and spotted onto an existing *Mucor* array growing on PCAMS agar and the resultant plate was screened for changes in the dispersal phenotype. A photograph of an example screen plate is on the right (photo by Scott Chimileski and used with permission). **d** Dispersal distances along *Mucor* hyphae of putative *Serratia* Tn5 mutants compared to the wild-type *Serratia*, after 7 days of growth. Mutants Tn5_11 and Tn5_54 killed the fungal host and are marked with red "X" over mycelia. One mutant, Tn5_13, displayed limited dispersal along the fungal hyphae. Bars are mean distance from the center of the spot to the bacterial colony edge (±1 standard deviation, *n* = 5). Dispersal assays were conducted on BHI agar. **e** Images of the wild-type *Serratia*, the dispersal deficient Tn5_13, and a killer mutant Tn5_11 with and without *Mucor* growing on PCAMS agar. The wild-type *Serratia* disperses to the edge of the hyphae, as outlined in red, while the dispersal deficient Tn5_13 does not disperse as far, as outlined in red. Mutant Tn5_11 completely inhibits the growth of *Mucor*. See Supplementary Fig. 6 for more genetic and phenotypic data from the selected transposon mutants

of the predicted chitinase genes in the *Serratia* genome when growing on chitin-rich fungal networks (Supplementary Data 1). Homologs of these genes have been associated with antifungal properties in other *Serratia* species[39,40] and downregulation of chitinases may partly explain why *Serratia* does not completely kill *Mucor*. Many of the genes with increased expression levels were associated with carbohydrate catabolism (Fig. 3b), again suggesting that growth on fungal networks alters the metabolism of *Serratia*.

A single time point of RNA-seq data cannot capture dynamic transcriptional responses that may occur during the different stages of microbial interactions. However, the overall pattern of limited differential expression of *S. proteamaculans* BW106 when grown with *Mucor* aligns with a previous RNA-seq study of the bacterium *Serratia plymuthica* grown in the presence of the fungus *Rhizoctonia solani*[41]. In that study, only 38 genes were differentially expressed, similar to the magnitude of differentially expressed genes observed in our study.

Transposon mutagenesis provides a complementary approach to RNA-seq by identifying specific genes necessary for dispersal on fungal hyphae. We used a Tn5 transposon mutagenesis system[42] to generate *Serratia* mutants that were then screened on arrays of *Mucor* networks (Fig. 3c). Using this approach, we initially identified 59 mutants that demonstrated altered colony appearance or dispersal phenotypes on *Mucor* networks, ranging from complete lack of dispersal on fungal hyphae to killing of the fungal host (Fig. 3d). These 59 mutants were further re-screened for fungal-mediated dispersal using the co-spotting assay described above, and six mutants with distinct phenotypes were selected for whole-genome sequencing to identify transposon insertion sites.

In our dispersal assay, the most striking mutant was Tn5_13, which was entirely dispersal deficient on both *Mucor* networks (Fig. 3d) and low-agar medium (Supplementary Fig. 5), suggesting a loss of motility. Using whole-genome sequencing, we discovered that the Tn5 transposon had disrupted the *fliS* gene in mutant Tn5_13 (Supplementary Fig. 6). FliS has not been well-characterized in *Serratia* species, but in other bacteria, FliS is a flagellin-specific chaperone that coordinates export of flagellin from the cell[43]. Disruption of this key regulator of flagellin biosynthesis leads to the production of short flagella and loss of motility in *Bacillus subtilis* and *Salmonella typhimurium*[44,45] and may play similar roles in *Serratia*. Over 40 genes are predicted to be involved with flagellar biosynthesis and regulation in the *S. proteamaculans* BW106 genome. Screening 6886 mutants provided ≈1× coverage of the predicted genes in the 5.6 Mb genome of *S. proteamaculans* BW106. More subtle loss-of-function flagellar mutants may have been difficult to identify using our macroscopic phenotypic approach. Despite this limitation, our screen supports previous targeted knockout studies and confirms a key role of flagella in fungal-mediated bacterial dispersal[18,36,46].

Other mutants were motile and did not display the same striking loss of dispersal on fungal networks as Tn5_13 (Supplementary Fig. 5), but they did display altered interaction outcomes or dispersal phenotypes (Supplementary Fig. 6) that provided further insights into other genes that may impact the outcomes of *Serratia* dispersal on *Mucor* networks. Surprisingly, mutants Tn5_11 and Tn5_54 completely killed *Mucor*, and thus *Serratia* did not have a fungal network present to facilitate dispersal across the agar surface (Fig. 3e, Supplementary Fig. 6). The Tn5 transposon inserted into a predicted ADP-heptose synthase in Tn5_11 and into a predicted ferric-binding periplasmic protein in the enterobactin operon in mutant Tn5_54. Why transposon insertions in these two genes caused *Serratia* to kill *Mucor* is unclear, but disruption of metabolic

pathways associated with these gene products may have resulted in the accumulation of metabolites with antifungal activity.

Three other mutants—Tn5_55, Tn5_57, and Tn5_59—formed *Serratia-Mucor* co-spots with altered colony edges or thicknesses (Supplementary Fig. 6), but the overall dispersal distance of these mutants on *Mucor* did not significantly change (Fig. 3d). These mutants had transposon insertions in genes related to phosphate metabolism (PhoU protein in Tn5_55)[47], a gene with an unknown function (in Tn5_57), and a gene known to be essential in phospholipid biosynthesis (a glycerol-3-phosphate dehydrogenase in Tn5_59)[48], suggesting that phosphate and phospholipid metabolism of *Serratia* can impact colony formation on *Mucor* networks.

To further investigate potential genetic mechanisms underlying *Serratia* dispersal on *Mucor* networks, we compared the genomes of the three closely related *S. proteamaculans* strains that showed different dispersal patterns on *Mucor*: BW106 and B-41162, which disperse on *Mucor*, and B-41156, which does not (Supplementary Fig. 2). Of the 62 gene annotations absent in the genome of B-41156, but present in BW106 and B-41162, one stood out: the gene *fliQ*, which is part of the *fliLMNOPQR* flagellar biosynthesis operon (Supplementary Data 2). In motile species of the Enterobacteriaceae, FliQ is one of the six transmembrane proteins that make up the flagellar export apparatus[49,50]. A frameshift deletion in *fliQ* of B-41156 leads to predicted loss of function (Supplementary Fig. 7). This loss of function was supported with a motility assay: while strains BW106 and B-41162 showed rapid dispersal across high motility (0.6% agar) plates (515% and 525% increase in growth, respectively), strain B-41156 showed a limited increase in growth (152%) (Supplementary Fig. 8). While other genomic differences could also contribute to the loss of spreading on fungal networks by B-41156, this observation reinforces the important role of flagella and motility in dispersal facilitation of bacteria by fungal networks.

**Fungal networks shape cheese rind microbiome composition.** Previous studies have described the potential existence of fungal-mediated bacterial dispersal in soil systems through pairwise interaction studies of a few laboratory strains[16,19,36]. Whether these interactions, which were studied in isolation, can shape the composition of multispecies communities has not been determined. Strong pairwise interactions may be dampened by multispecies interactions that can occur in communities with three or more species[51]. Given our observation that motility is required for *Serratia* to disperse on fungal networks, we predicted that fungal-mediated bacterial dispersal would be unevenly distributed across cheese rind bacteria: other motile Proteobacteria species would disperse on fungal networks while non-motile Actinobacteria and Firmicutes species would have limited dispersal on fungal networks. We also predicted that this uneven dispersal facilitation would have consequences for the assembly of multispecies communities, with motile Proteobacteria favored over other bacterial taxa in communities when dispersal-promoting fungal networks were present.

Using the same co-spot approach described above with *Serratia*, we screened 22 cheese rind bacterial isolates spanning 13 bacterial genera and three phyla (Proteobacteria, Firmicutes, and Actinobacteria) for their ability to disperse on fungal networks. The genera of Proteobacteria screened, including *Hafnia*, *Halomonas*, *Providencia*, *Pseudomonas*, *Psychrobacter*, and *Vibrio*, generally have actively motile cells that swim or swarm using flagella[52–57], while the Firmicutes and Actinobacteria genera (*Staphylococcus*, *Arthrobacter*, *Brevibacterium*, *Brachybacterium*, *Corynebacterium*, *Leucobacter*, and

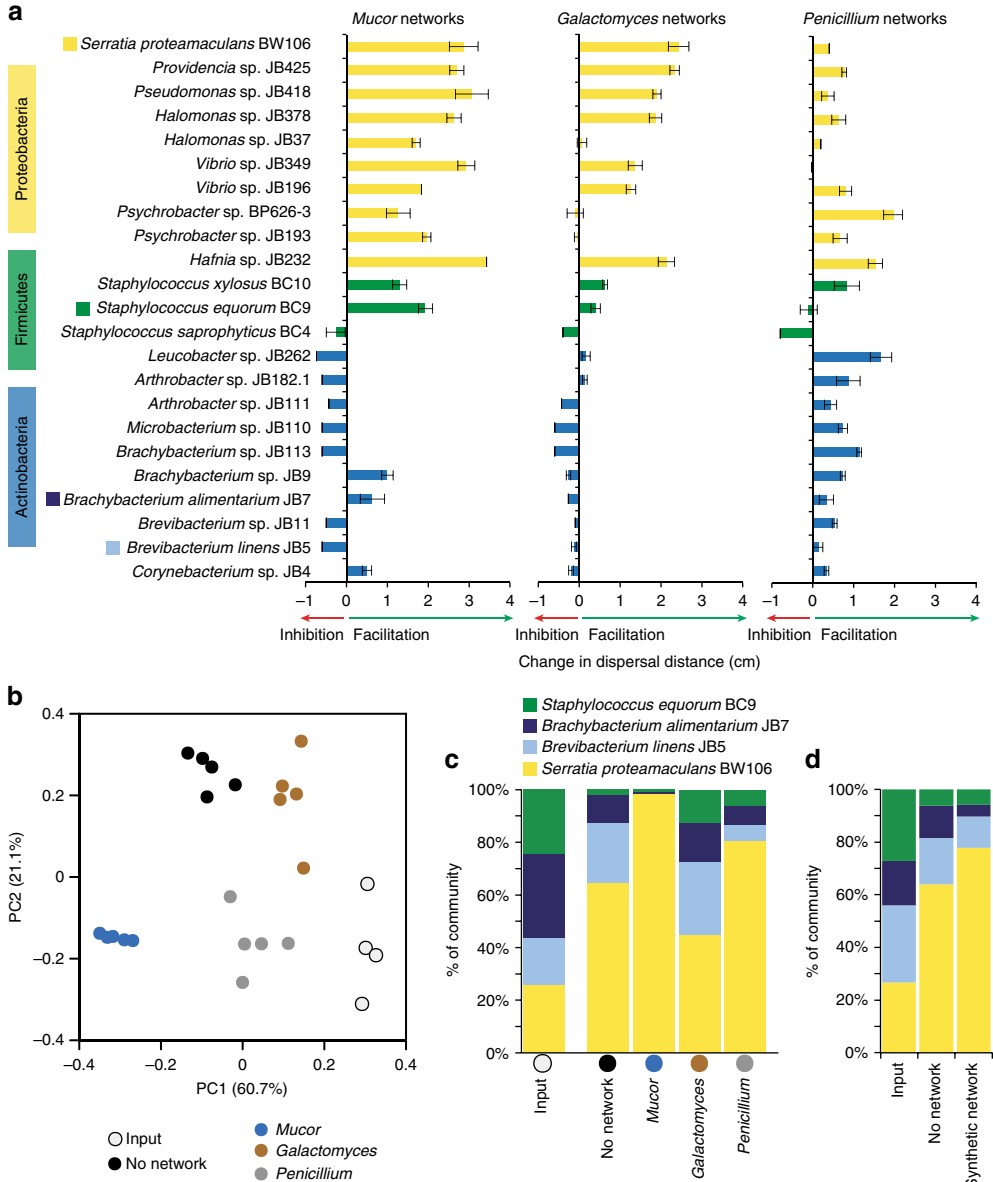

**Fig. 4** Fungal networks facilitate the dispersal of Proteobacteria and shape the diversity of cheese rind communities. **a** Dispersal of various cheese rind bacteria on fungal networks. Data show change in dispersal distance (as measured from the center of the bacterial colony to the edge) on *Mucor*, *Galactomyces*, or *Penicillium* networks compared to growth alone after 14 days of growth. Dispersal assays were conducted on BHI agar. Bars are mean ±1 standard deviation ($n = 3$). Taxa indicated with a colored box were used in community experiments. **b** Principal coordinates analysis of relative abundance data from experimental communities on cheese curd agar with different fungal networks. "Input" indicates the input inoculum used to set up the experiment. The "No network" treatment included no fungus. *Mucor*, *Galactomyces*, and *Penicillium* treatments included living fungal networks of the respective fungi. Each community treatment was replicated five times. The addition of fungal networks shifted the composition of the bacterial communities compared to no network communities (PERMANOVA $F = 72.14$, $p < 0.001$). **c** Mean relative abundance (% of total CFUs) of bacteria within the experimental communities with different fungal networks after 14 days of growth. Treatments are the same as in **b**. **d** Mean relative abundance (% of total CFUs) of bacteria in control communities ("No network") and communities with a glass fiber network ("Synthetic network") after 14 days of growth. Synthetic networks shifted bacterial community composition compared to the no network controls (PERMANOVA $F = 7.38$, $p < 0.01$)

*Microbacterium*) are considered non-motile[58–63]. Fungal networks strongly facilitated the dispersal of Proteobacteria compared to Firmicutes and Actinobacteria on *Mucor* (nested ANOVA $F_{2,65} = 13.80$, $p < 0.001$; Fig. 4a) and *Galactomyces* networks ($F_{2,65} = 9.94$, $p < 0.01$; Fig. 4a), but there were no differences across phyla on *Penicillium* networks ($F_{2,65} = 2.97$, $p = 0.074$; Fig. 4a) networks. This pattern of differential dispersal facilitation across bacterial phyla strongly supported our prediction that most motile Proteobacteria can disperse on fungal networks, while Firmicutes and Actinobacteria cannot.

To test whether fungal networks can impact cheese rind microbiome diversity by promoting the growth of Proteobacteria, we inoculated CCA with equal CFUs of *S. proteamaculans* BW106 (Proteobacteria—high dispersal on fungal networks), *Staphylococcus equorum* BC9 (Firmicutes—medium dispersal on fungal networks), *Brevibacterium linens* JB5 (Actinobacteria—low dispersal on fungal networks), and *Brachybacterium alimentarium* JB7 (Actinobacteria—low dispersal on fungal networks) as well as the yeast *Debaryomyces hansenii*. The yeast was included because it is a common component of cheese rind microbial

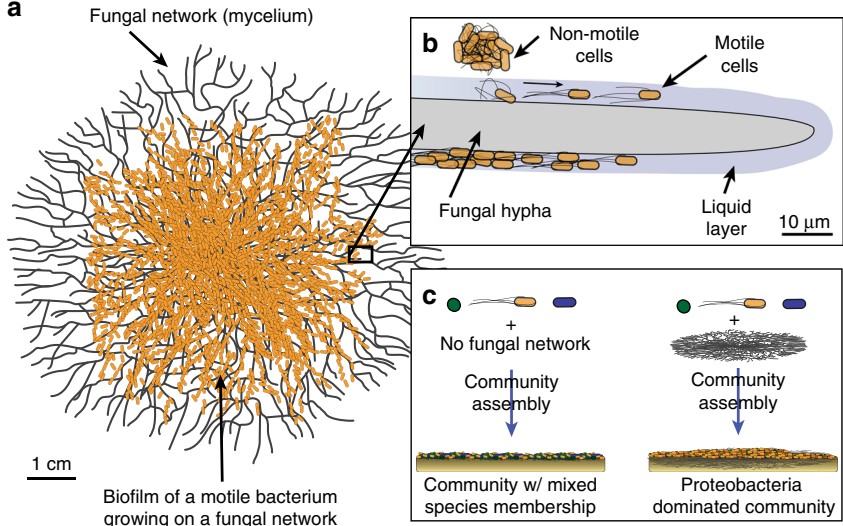

**Fig. 5** Overview of mechanisms and consequences of fungal-mediated bacterial dispersal in cheese rind microbiomes. **a** Cells of motile Proteobacteria spread across fungal networks, leading to increased dispersal across cheese surfaces. **b** *Serratia proteamaculans*, and likely other motile Proteobacteria, use flagella to swim in the liquid layers surrounding *Mucor* hyphae. **c** The presence of fungal networks in cheese rind communities leads to the promotion of motile Proteobacteria growth over non-motile community members

communities and it facilitates the deacidification of the cheese curd medium[27]. Four different fungal treatments were then applied to these experimental communities: (1) No network, where no fungus was inoculated, (2) *Mucor*, (3) *Galactomyces*, and (4) *Penicillium*. Fungi were added as spores and were allowed to form mycelia as they grew with the bacterial communities. After 2 weeks of growth, a rind had formed on the cheese surface and the experimental communities were harvested to determine CFUs of each bacterium present.

As predicted, the addition of fungal networks shifted the composition of the bacterial communities compared to no network communities (PERMANOVA $F = 72.14$, $p < 0.001$; Fig. 4b), with *Mucor* communities having the highest relative abundances of *Serratia* across all treatments (No network: 64.9 ± 2.1%; *Mucor*: 98.5 ± 0.3%; *Galactomyces*: 45.2 ± 3.1%; *Penicillium*: 80.1 ± 3.0%; ANOVA $F_{3,19} = 89.2$, $p < 0.001$; Fig. 4c). The limited impact of *Galactomyces* on community composition is surprising, given that this fungus strongly promoted the dispersal of *Serratia* alone (Fig. 4a), and indicates that pairwise interactions cannot always predict outcomes in multispecies communities. Surprisingly, *Penicillium* caused an increase in the relative abundance of *Serratia* in the experimental communities even though it demonstrated limited dispersal facilitation in co-culture experiments. Previous work in this system demonstrated that *Penicillium* inhibits the growth of *Brevibacterium* and *Brachybacterium* to a greater extent than it does *Staphylococcus* or *Serratia*, possibly due to the production of antibacterial compounds[27], and this differential inhibition may be driving the shift in bacterial composition when *Penicillium* is present.

To tease apart the relative impact of dispersal of bacteria on networks versus other abiotic or biotic interaction mechanisms, we repeated the same experiment but instead added glass fiber networks described above to create synthetic networks on the cheese surface (Supplementary Fig. 9). We predicted that the relative abundance of *Serratia* would increase in the presence of synthetic networks as we observed in the *Mucor* treatments. After 2 weeks of growth, the synthetic networks had shifted in composition (PERMANOVA $F = 7.38$, $p < 0.01$; Fig. 4d), with significantly higher relative abundance of *Serratia* compared to the no network control treatment (ANOVA $F_{1,13} = 9.25$, $p < 0.01$; No network: 64.1 ± 9.9%; Synthetic network: 77.8 ± 6.6%). The

more limited effects of the synthetic networks on community composition compared to living *Mucor* networks (Fig. 4c) could be because synthetic networks do not grow with the bacterial populations over time and/or because biotic cues or conditions generated by the fungi are missing. Regardless, these data demonstrate that the presence of physical networks alone can cause bacterial communities to shift in composition through differential dispersal facilitation of bacterial species.

## Discussion

Fungi and bacteria co-occur in many types of microbiomes, including animal hosts, soils, and food systems[17]. Despite the potential for substantial diversity of bacterial–fungal interactions in such microbiomes, the biology of most bacterial species has been studied without considering the potential role of fungi as mediators of evolutionary and ecological processes. We demonstrate that strong, pairwise interactions between bacteria and fungi can not only shape the small-scale dispersal dynamics of a single bacterial species, but can also impact the diversity of multispecies bacterial communities (Fig. 5). The results presented here—in concert with recent studies from other systems[28,64–66]—suggest that a thorough understanding of mechanisms regulating microbiome assembly requires both eukaryotic and prokaryotic perspectives.

It has been previously demonstrated that motile bacteria can migrate on fungal hyphae[16,18,20,21,36], but studies of the genetic mechanisms of these interactions are limited and are often based on the selection of a few candidate genes[36]. Our novel interaction-based transposon mutagenesis screen used an untargeted approach to identify any non-essential genes controlling this interaction. We determined that flagella-mediated motility is essential for the dispersal of *S. proteamaculans* in liquid layers on fungal hyphae (Fig. 5b). Previous studies of fungal-mediated bacterial dispersal have not tested the effect of these pairwise interactions in the context of multispecies communities. Using the tractable cheese rind model microbiome[25,27,28], we demonstrated that fungal-mediated bacterial dispersal can shape the composition of relatively simple multispecies microbiomes by promoting the growth of motile Proteobacteria (Fig. 5c). This dispersal facilitation was not limited to a single strain of

bacterium, but was also identified in numerous strains of *S. proteamaculans* and *Serratia liquefaciens* as well as a range of other Gammaproteobacteria species. We acknowledge that our cheese rind microbial communities are relatively low in species diversity, and the impacts of fungal-mediated bacterial dispersal interactions may be more diffuse in complex communities. But given that *Mucor* and other filamentous fungi often co-occur in environments with motile Proteobacteria[67–69], we predict that this biophysical interaction can play key roles in determining the composition of other microbiomes.

Many aspects of the diversity, mechanisms, and impacts of these interactions remain to be explored. How do co-occurring motile bacteria interact and compete on fungal networks? Do motile pathogenic species, such as *Listeria monocytogenes*, use fungal networks to disperse across cheese surfaces and other environments? Given the high cost of producing flagella[18], can fungal networks impact the evolution of motility traits in bacteria? Future work using the cheese rind model microbiome and other tractable systems will continue to explore the causes and consequences of fungal-mediated bacterial dispersal.

## Methods

**Isolation and maintenance of cultures**. Strains were isolated from the rinds of cheeses by serially diluting cheese rind scrapings on plate count agar with milk and salt, or PCAMS[27]. For all experiments described below, strain BW106 of *S. proteamaculans* isolated from the Saint-Nectaire cheese described above was used with strain SN1 of *M. lanceolatus* (isolated from the same cheese). Inocula for all experiments were created from frozen glycerol stocks of bacterial overnight cultures grown in BHI broth for 16 h. These experimental glycerol stocks were plated to determine CFUs per µL of inoculum, and these CFU densities were used to standardize the inputs into the experiments below. Fungal stocks were created by either scraping the surface of a plate containing spores (*Mucor* and *Penicillium*) or from liquid overnight cultures grown in yeast peptone dextrose broth (*Galactomyces*).

To comprehensively identify the species of *Serratia* and *Mucor* isolated from Saint-Nectaire, as well as *Serratia* strains received from the United States Department of Agriculture ARS Culture Collection (NRRL), we used whole-genome sequencing to create draft genomes as previously described[28]. Draft genome sequences have been deposited in NCBI (see Supplementary Table 1 for accession info). To construct a phylogenomic tree of the *Serratia* species, we first identified single-copy genes shared by all *Serratia* species from RAST-annotated[70] genomes that were assembled using CLC Genomic Workbench. An alignment of all single-copy genes was made using MUSCLE[71], and then a maximum likelihood tree was constructed using RAxML[72] using the General Time Reversible + Gamma (GTRGAMMA) model. To place the *Mucor* strain SN1 isolated from cheese within a phylogenetic context, we used previously published 18S rRNA, 28S rRNA, ITS, and rpb1 sequences from *Mucor* isolated from cheese and other environments[73]. Using a low coverage (10×) assembly of reads from a 100-bp, paired-end, genomic library of Mucor strain SN1 (Supplementary Table 1), we extracted 18S rRNA, 28S rRNA, ITS, and rpb1 genes for this phylogeny. An alignment of concatenated sequences from representative strains was used to construct a maximum likelihood phylogeny using RAxML (Supplementary Fig. 10).

**Network dispersal co-spot assays**. To measure the dispersal of *Serratia* on actively growing fungal networks, 250 CFUs of *Serratia* were inoculated in 10 µL of 1× phosphate buffered saline (PBS) onto the center of a BHI agar (1.5% agar) plate with 500 CFUs of a fungus (*Mucor*, *Galactomyces*, and *Penicillium* networks) or with no fungus (No network). Bacterial and fungal cells used in the co-spots were from frozen glycerol stocks with a known number of CFUs/µL. The 10-µL spot of PBS dried within a few minutes of adding it to the plate and did not impact the growth or dispersal of the bacterial cells. We used an initial ratio of 1:2 *Serratia* to *Mucor* based on pilot experiments where this ratio gave most consistent outputs and because final bacterial and fungal densities using this approach reached levels similar to those found in cheese rinds[74,75]. Plates were incubated at 24 °C for 14 days.

In addition to BHI agar, various other types of media were used to demonstrate that this interaction is not specific to one medium type (Supplementary Fig. 1). These media included CCA[27], PCAMS[27], potato dextrose agar (PDA), and yeast extract sucrose agar (YES). Most experiments in the manuscript were conducted on BHI, PCAMS, or CCA. Because the hydration level of different types of media can impact the motility of bacteria[18,23], we used media that had similar water activity ($a_w$) to what is found in fresh cheese curds (BHI $a_w = 0.992$; PCAMS $a_w = 0.992$; CCA $a_w = 0.972$; fresh cheese curds $a_w = 0.988$).

To quantify the extent of bacterial dispersal across the fungal network, we developed a transect "tap and transfer" method. A sterile toothpick, measuring the

radius of the Petri dish (4.2 cm), was tapped on the fungal network from the center of the plate (center of the initial spot of bacterial and fungal cells) to the outside edge of the actively growing fungal mycelium. The toothpick was then removed from the experimental plate and tapped onto the surface of a new BHI agar plate containing cycloheximide (50 mg/mL) to inhibit fungal growth for *Penicillium* and *Mucor* networks and natamycin (21.6 mg/L) for *Galactomyces*, which is resistant to cycloheximide. Transfer plates were incubated for 24 h at 24 °C and the length of the transects with *Serratia* growth was used to infer the distance traveled across the fungal network. ANOVAs were used to determine significant differences in dispersal distance across the fungal network treatments.

Synthetic fungal networks, made of glass fibers, were used to determine if active motility was necessary for dispersal on fungal networks. After a 10-µL spot of *Serratia* inoculum in PBS containing 250 CFUs absorbed into BHI agar, sterile glass fibers were placed on the surface of the agar at a similar density to living fungal networks. Glass fibers have a similar thickness (8 µm) to that of fungal hyphae, which range from 4 (*Penicillium*) to 25 µm (*Mucor*) in diameter[73,76]. After 2 days of growth, the presence of dispersal across the synthetic network was noted and representative plates were photographed.

To determine if *S. proteamaculans* could spread on existing fungal networks, we inserted a plasmid (pGLO, BioRad) containing a gene for GFP with an *araC* promoter into *S. proteamaculans* BW106. Electrocompetent cells of *S. proteamaculans* were made by first growing up overnight cultures for 16 h in liquid BHI medium. The cultures were then diluted 100-fold in liquid BHI medium in a baffle flask. The cultures were grown with agitation for 2–4 h until the $OD_{600}$ measured 0.5. The culture was chilled on ice for 15 min, then centrifuged at 3000×g and 4 °C for 10 min. The resulting pellet was washed four times at decreasing volumes (125, 75, 25, and 5 mL) of 10% glycerol. After the final wash, the cells were re-suspended in 2.5 mL 10% glycerol and frozen at −80 °C for at least 18 h. These cells were used for electroporation at 1.8 kV, 25 µF, and 200 Ω, to transform the strain with the pGLO plasmid.

Square Petri dishes (12 cm × 12 cm) containing PCAMS with ampicillin (100 mg/mL) and arabinose (2 mg/mL) were inoculated with 10,000 CFUs of either *Mucor*, *Galactomyces*, or *Penicillium*. Plates were incubated at 24 °C for 48 h before inoculum of *S. proteamaculans* BW106 containing the pGLO plasmid was tapped into the top left corner of the plate using the tip of a sterile toothpick. Each fungal network treatment and a control treatment (No network) were replicated four times. After 48 h of growth when the fungal hyphae had formed a complete lawn across the plates, the area colonized by *Serratia* was determined by photographing plates while exposed to a long-wave UV lamp. ImageJ was used to trace outlines of the area colonized by *Serratia* and to quantify the total area colonized. ANOVAs were used to determine significant differences in area colonized across the fungal network treatments.

**Co-culture growth assays**. *Mucor* growth was quantified by co-inoculating *Mucor* (500 CFUs) and *Serratia* (250 CFUs) at the center of a Petri dish. Controls of *Mucor* (500 CFUs) in 10 µL PBS and *Serratia* (250 CFUs) in 10 µL PBS were also created. Each treatment was replicated five times each for two time points: 7 and 14 days. At each time point, the population was quantified using a whole plate harvest technique. With a sterile pipette tip, the agar and microbes were excised from the plastic Petri dish and placed into a 710-mL Whirl-Pak® bag with 30 mL of 1× PBS. The agar and microbes were manually homogenized by rolling a closed 50-mL conical tube across the outside of the bag to pulverize and mix the agar with the PBS. The homogenate was serially diluted on selective media (BHI agar with cycloheximide 100 µg/mL for bacteria and BHI agar with chloramphenicol 50 µg/mL for fungi). The CFUs on these plates were counted to quantify the abundance of *Serratia* and *Mucor*. We chose CFU quantification and not other measures of bacterial and fungal abundance (biomass) because it is a commonly used measure of bacterial and fungal abundance in cheese rinds and systems with easy to culture microbes, it is a metric that can be easily used across both bacteria and fungi, and it captures the total number of reproductive units in both types of microbes. We acknowledge that in *Mucor*, CFUs can originate from both hyphal fragments and spores.

**Microscopy**. Thin layers (≈0.7 mm in height) of BHI agar (1.5% agar) were poured into 60 mm × 15 mm Petri dishes on a level surface and left to set for 4 h. *Serratia* and *Mucor* were co-inoculated in the center of the Petri dish in a 5-µL spot of liquid 1× PBS containing 115 CFUs/µL of *Serratia* and 1150 CFUs/µL of *Mucor*. This ratio of *Serratia* and *Mucor* cells provided an ideal distribution of *Serratia* and *Mucor* colonies for imaging initial contacts between the bacterium and fungus. The Petri dishes were incubated at room temperature (24 °C) in the light for at least 10 h prior to imaging. Images and movies of *Serratia*–*Mucor* interactions were taken on a Nikon TiE inverted microscope using phase contrast imaging with an Andor Zyla 5.5 camera under 40× magnification (0.6 NA). Three replicate *Serratia*–*Mucor* contacts were imaged on multiple days to determine the robustness of phases of cell contact and growth. Three replicate time-lapse examples are presented in the manuscript.

**RNA-seq**. Experimental populations of *S. proteamaculans* BW106 growing alone (No network) and on *Mucor* networks (+ *Mucor* network) were constructed by inoculating circular 100-mm Petri dishes containing 20 mL of PCAMS[27]. Plates

were inoculated with 5000 CFUs of *S. proteamaculans* (No network) or a mix of 5000 CFUs of *S. proteamaculans* and 5000 CFUs of *Mucor* (+ *Mucor* network) and incubated at 24 °C for 27 h. Each treatment was replicated three times. Cells were harvested by scraping the agar surface with a sterile razor blade to remove most of the microbial biomass. Harvested cells were stored in RNAProtect Reagent (Qiagen) to stabilize mRNA and frozen at −80 °C. RNA was extracted using a standard phenol–chloroform protocol described previously[77]. This protocol uses a standard bead-beating step in a lysis buffer to release cell contents from pelleted cells in RNAProtect. To ensure that the RNA was of high quality and not degraded, 500 ng of each RNA prep was run and visualized on a 1.5% agarose gel. DNA was removed from the nucleic acid pool using a TURBO DNA-free kit (Life Technologies). 5S rRNA and tRNA were depleted using MEGAClear (Life Technologies) kits. 16S and 23S rRNA were depleted using RiboZero (Illumina) kits. To remove both fungal and bacterial ribosomal RNA, yeast and bacterial rRNA probes from the RiboZero kits were mixed 1:2 and used for rRNA depletion. To confirm that the samples were free of DNA contaminants, a PCR of the 16S rRNA was conducted with standard 16S primers (27f and 1492r).

RNA-seq libraries were constructed from purified mRNA using the NEBNext Ultra RNA Library Prep Kit for Illumina (New England Biolabs) using manufacturer's instructions and sequenced using paired-end 100 bp reads on an Illumina HiSeq Rapid Run by the Harvard Bauer Core Facility. About 16 million reads were sequenced for each library. Only forward reads of the paired ends were used for analysis. Raw reads and differential expression data have been submitted to the NCBI GEO database. The project is available as GEO Series Accession Number GSE85095. Analysis of RNA-seq libraries, including read mapping, normalization, and quantification of transcript abundances, was done using Rockhopper version 1.3.0[78] with default settings as previously described[28]. The *S. proteamaculans* BW106 genome was concatenated and used as the reference genome for read mapping. It has been deposited at DDBJ/ENA/GenBank under the accession MCGS00000000. Expression values were normalized by the upper quartile of gene expression. We considered differentially expressed genes to be those that were greater than 2-fold change in expression when comparing No network to + *Mucor* network replicates. Expression differences were deemed statistically significant based on a *q*-value of <0.05. Rockhopper's *q*-values are *p*-values adjusted for false discovery rate using the Benjamini–Hochberg procedure. Functional assignment of differentially expressed genes was determined from the SEED subsystem annotations from the RAST-annotated[70] genome of *S. proteamaculans* BW106.

**Transposon mutagenesis.** To identify genes that impact *Serratia* dispersal on fungal networks, we used the EZ-Tn5™ <KAN-2>Tnp Transposome™ Kit (Epicentre) to generate a transposon mutant library. Electrocompetent cells (as described above) were used to electroporate the EZ-Tn5™ Transposome into *Serratia*. Cells were plated onto PCAMS containing 50 µg/mL kanamycin to select for successful transformants. Colonies were then patched onto Nunc Omni Trays in an 8 × 12 grid to match a 96-pin replicator.

To screen the mutant library for altered dispersal phenotypes on fungal networks, each 96-array of mutants was tapped onto an array of *Mucor* networks that had been growing for 24 h. *Mucor* networks were generated by tapping the 96-pin replicator onto a lawn of three-day-old *Mucor* to collect spores, which were then transferred to a fresh Omniplate containing PCAMS with 50 µg/mL kanamycin. This array of 96 spots of *Mucor* was incubated at 24 °C for 24 h, after which a 96-array of *S. proteamaculans* mutants was inoculated on the networks using the same 96-pin method. After another 24 h of growth, the resulting co-cultures were visually screened using 4× magnification under a dissecting microscope. We confirmed that lack of dispersal was not the result of poor growth by also tapping the mutant library on Omni Trays without *Mucor*. A subset of putative mutants was isolated and rescreened using the co-spot assay described above in "Network dispersal co-spot assays".

Transposon insertion sites were determined by using whole-genome sequencing of the putative mutants. DNA was extracted using MoBio PowerSoil DNA extraction kits from streaks generated from a single bacterial colony grown for 2–3 days on BHI agar. Approximately 1 µg of purified gDNA was sheared using NEBNext dsDNA fragmentase (New England Biolabs) to a size range of approximately 300–700 bp NEBNext Ultra DNA Library Prep Kit for Illumina (New England Biolabs). Libraries were spread across multiple sequencing lanes with other projects and were sequenced using 100-bp, paired-end reads on an Illumina HiSeq 2500. Approximately 10 million reads were sequenced for each genome. Failed reads were removed from the libraries and reads were trimmed to remove low-quality bases and were assembled to create draft genomes using the de novo assembler in CLC Genomics Workbench 8.0. Insertion sites were identified by generating a BLAST searchable database of the whole genome and then searching for the transposon sequence.

Putative mutants were also screened for loss of motility by spotting 250 CFUs of cells in 10 µL of 1× PBS in the center of a Petri dish. Each mutant was screened for motility using five biological replicates. Plates were incubated at 24 °C, and the radius of bacterial dispersal was measured after 14 days of growth.

**Comparative genomics.** To identify genes that might be absent in dispersal deficient *S. proteamaculans* strain B-41156, we compared the presence of predicted

protein coding genes of that strain to two closely related motile strains, BW106 and B-41162. Core and accessory genes were identified using PGAP[79], as previously described for *Staphylococcus* species[28]. Species-to-species orthologs were identified by pairwise strain comparison using BLAST with PGAP defaults: a minimum local coverage of 25% of the longer group and a global match of no less than 50% of the longer group, a minimum score value of 50, and a maximum E value of 1e−8. Multistrain orthologs were then found using MultiParanoid in PGAP.

**Community experiment.** To determine how fungal networks might impact the dispersal of other bacterial taxa that live in cheese rinds, the same network dispersal co-spot assay described above for *S. proteamaculans* BW106 was used to measure fungal-mediated bacterial dispersal of an additional 22 bacterial isolates spanning the phyla Proteobacteria, Firmicutes, and Actinobacteria (Supplementary Table 1). These bacteria are commonly the dominant species in rind communities found on Saint-Nectaire cheese and other natural and washed rind cheeses[27]. As with co-spot experiments above, bacteria were spotted alone or with each of the three fungal networks (*Mucor*, *Galactomyces*, and *Penicillium*). Based on low variation observed across biological replicates in pilot experiments, three biological replicates were conducted for each treatment. The dispersal distance from the center of the co-spot to the edge of the bacterial colony was measured using the same transect and tap method described above. Data are expressed as change in dispersal distance when on a network compared to alone. Nested ANOVAs were used to test for significant differences in dispersal distances on fungal networks between the phyla Proteobacteria, Firmicutes, and Actinobacteria.

An in vitro community reconstruction approach was used to determine how fungal networks impact bacterial community composition. CCA was used to grow a model cheese rind community consisting of *B. linens* JB5, *B. alimentarium* JB7, *S. proteamaculans* BW106, and *S. equorum* BC9 (see Supplementary Table 1 for strain origin information). The yeast *D. hansenii* 135B was also added to these communities because it is widespread in cheese rinds and facilitates deacidification of the cheese curd[27]. Three fungal networks treatments were applied to this community: *Mucor*, *Galactomyces*, or *Penicillium*. As with co-spots above, bacterial cells and fungal spores used in this experiment came from −80 °C glycerol stocks with a known number of CFUs/µL. All species were mixed in approximately equal concentrations ("Input") for a total CFU of 20,000. A control community with no fungal network ("No network") was also created by adding the same volume of 1× PBS used for the fungal network inocula. The communities were plated onto the surface of 20 mL of CCA dispensed in 100 mm Petri dishes. During this experiment, fungal networks grew with the bacterial cells on the surface of the cheese curd to form a rind. Experimental units were incubated in the dark for 4 days at 24 °C and then for 10 days at 14 °C in order to mimic the conditions of a cheese cave. After 2 weeks, each community was homogenized in 1× PBS, serially diluted, and then plated onto PCAMS media for the quantification of each species. All four bacteria have unique colony morphologies and pigments (Supplementary Fig. 11) and can be easily distinguished from one another.

To isolate the effect of physical networks for growth and dispersal of bacterial cells from other chemical or biological effects of the fungal networks, we repeated the experiment with a synthetic glass fiber network treatment. The same input inoculum described above containing *B. linens* JB5, *B. alimentarium* JB7, *S. proteamaculans* BW106, *S. equorum* BC9, and the yeast *D. hansenii* 135B was added to 14 CCA plates. To seven replicate plates, 1 g of the synthetic glass fiber network described above was spread across the surface of the agar after the inoculum had dried onto the cheese curd surface ("Synthetic network", Supplementary Fig. 9). The remaining seven replicate plates were not manipulated and served as controls ("No network"). Experimental units were incubated in the dark for 4 days at 24 °C and then at 14 °C for 10 days as above. Bacterial community composition was determined by counting colonies as described above.

In both the fungal network and synthetic network experiments, the differences in community composition across treatments were determined with PERMANOVA and changes in the relative abundance of *Serratia* across treatments were determined using ANOVA. Principal coordinates analysis was used to visualize differences in community composition across replicates using the Bray-Curtis dissimilarity index.

**Data availability.** Draft genome sequences for *Serratia* and *Mucor* strains have been deposited in the NCBI BioSample database (accession codes are listed in Supplementary Table 1). The *S. proteamaculans* BW106 draft genome has been deposited in the DDBJ/ENA/GenBank databases under accession code MCGS00000000. RNA-seq transcriptomic data have been deposited in the NCBI Gene Expression Omnibus (GEO) database under accession code GSE85095. All other relevant data are included in this published article and its Supplementary Information files, or can be obtained from the corresponding author upon request.

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

## Acknowledgements

This work was supported by the Tufts Collaborates Seed Grant Program from the Tufts University Office of the Provost and Senior Vice President (to B.E.W. and J.S.G.), the Knez Family Fund (to B.E.W.), the Tufts Russell L. Carpenter Fund (to Y.Z.), USDA NIFA grant 2017-67013-26520 (to B.E.W.), NSF grant 1715552 (to B.E.W.), NSF grant CAREER-1554095 (to J.S.G.), and NSF grant CBET-1511340 (to J.S.G.). We thank Esther Miller, Elizabeth Landis, Brittany Niccum, Fredrick Lee, Jason Shaffer, and Lucas Brown for providing feedback on earlier versions of this manuscript.

## Author contributions

B.E.W. initiated, conceived, and supervised the project. J.S.G. initiated and supervised the cellular imaging. Y.Z. conducted all the experiments. E.K. performed the comparative genomics and other bioinformatic analyses in the paper. B.E.W., J.S.G., E.K., and Y.Z. wrote the paper.

## Additional information

**Competing interests:** The authors declare no competing financial interests.

