## [Peer Review File · Nature Communications]

Reviewer #1 (Remarks to the Author):

The MS by Zhang et al investigates the effect of three different fungal networks first on the dispersal of *Serratia proteamaculans*, and afterwards in the dispersal of an assemblage of a few bacterial strains including the same strain indicated previously.

The manuscript presents results from different types of experiments, including the comparison of dispersal on biotic/abiotic networks, mutagenesis and transcriptomics, and manipulation of (simplified) bacterial assemblages. Although this is highly commendable, the message of the manuscript is unfortunately not clear. In addition, the conclusion in terms of the demonstration of a community-level effect of fungal dispersal networks is (in the opinion of this reviewer) an overstatement. One fungal species and a handful of bacterial strains do not (by far) make a microbial community (although I acknowledge that this is more complex than two-species interactions). This point is one of the main aspects put to the fore in the introduction to the manuscript, but I consider that the authors should be more critical.

In addition, it is difficult to form a clear idea of the hydration level of the different media used (with a few exceptions). Additional information is required here, as this abiotic factor is very important when evaluating the relative contribution of fungus-facilitated dispersal versus colony expansion (which appears in the case of the strain used to play a very significant role on dispersal).

In the case of Figure 1, it is unclear which medium was used for the observation of the "unusual streaming colonies" (parts b and C). The same applies for figure d. In this case, indicating the hydration level of the medium is very important. In addition, if information concerning the matrix potential of the cheese rind is available, this should be included.

Supplementary Figure 1 is not related to the information given in the text. The figure presents the phylogeny of the strains isolated, not the formation of bacterial streams. Please refer to this supplementary figure in the material and methods.

For figure 2 one has to understand from the description of the methods that the fungi were inoculated as spores (is this right?) and simultaneously with the bacterial cells (which supposes the use of a liquid substrate to place both in the center of the Petri dish and thus a boost on bacterial dispersal by colony expansion; see for example comparison of figure 1a with 1c, which precisely correspond to a pre-form dispersal network). In this experiment one has to control for the effect of the bacterial on fungal germination and the density of the network between the different species. This is partially considered for what the authors call a static network. However, in the images the extent of the fungal network in each of the treatments is not indicated (using the same timing is not necessarily relevant and comparable in terms of the biology of the fungal species used). This needs to be indicated. One aspect that surprises me is the effect of *Serratia* on sporulation. I am quite surprised at observing no spores in either the *Mucor* or the *Penicillium* systems. Could the authors indicate their observations (for example see supplementary Figure 4)? This is particularly relevant when the authors discuss the effect on fitness based on CFU counting, and the apparent inhibitory effect discussed in the text. Please consider including values of total active biomass as a parallel measure to fitness.

The experiment attempting to identify mechanistic elements of the interaction through mutagenesis is particularly interesting, however, the results presented are disappointing. The authors focus on only one of the mutants. The reader wonders what are the genetic modifications of the other mutants with a modified phenotype. In addition, when referring to the role of flagella in dispersal, please use all the existing references in the topics (this is something already established by other groups in the past).

The experimental set-up of the experiments with artificial communities is unclear. Were the

bacterial strains co-inoculated in a static or in an active network? In addition, it is unclear how the relative abundance of each bacterial strain was measured. Was this based on differences in colony morphology? Please clarify all these elements. In addition, please add a control for the artificial communities without dispersal network in order to assess the effect of competition regardless of dispersal (your strains are likely to all have different growth rates, which could alter the ratios obtained regardless of dispersal). The bar corresponding to input is not defined.

Finally, the authors should revise all figure legends (including the supplementary) for elements missing in the description. These include: media used, undefined terms (e.g. N.D. SFig 5; -GW/+GW SFig 7), and scales for the images.

Reviewer #2 (Remarks to the Author):

Zhang et al. analyze the effect of fungal mycelia on the dispersal, gene expression and community composition of selected bacteria isolated from cheese rinds. In synthetic microcosms they reveal that mycelia allow for and differentiate bacterial dispersal and growth of bacteria and thereby enable temporal shifts of an (artificial) bacterial community. By performing RNA-seq., transposon mutagenesis and comparative genomic analysis they find that the presence of *Mucor lanceolatus* may not influence global gene expression of flagellar biosynthesis (or other genes/regulators) yet rather lead to upregulation of genes related to metabolic processes.

The authors thereby confirm many previously known concepts/facets of bacterial-fungal interactions (BFI) and show that many BFI also may take place in the microbial habitat of cheese rinds: e.g. i) the role of mycelia (and mycelial properties on bacterial dispersal and growth of both synthetic and more complex microbial communities, ii) the influence of bacterial motility on dispersal, iii) the presence of metabolic interactions of bacteria and fungi in the hyphosphere, iv) analyses of complex bacterial fungal communities (including molecular and genetic analyses), etc. Many of the claims of novelty of the manuscript hence may be overstatements.

Interesting and novel insights give the study's RNA-seq, transposon mutagenesis and comparative genomic analysis. These point at a downregulation of the chitinase genes and possible metabolic interactions of the *Serratia* with the fungus, rather than at an upregulation of dispersal-related genes. Although following a promising approach the given experiments seem to open more questions than answers; relevant additional experiments are missing to give transferable insights to the BFI ecology: e.g. to which degree are the changes due to altered nutrient status of the bacteria (i.e. independent of the presence of a fungus)? Do similar changes also occur in presence of other (dispersal promoting; not dispersal promoting) mycelia? To which degree are such up-/down-regulation dependent on the age of the fungal mycelia (cf. data in Fig. 2f)?

Similar resolution problems may also be observed in the experiments using synthetic multispecies microbiomes. While following an interesting approach, experimental data unfortunately does not allow discriminating the influence of dispersal from the influence of growth (only relative data of the biomass and no spatio-temporal resolution of the microbiome are provided; a control with artificial networks allowing for spatiotemporal distribution in the absence of bacterial fungal interactions (BFI) is missing). The data rather remain descriptive lacking in depth ecological insights beyond the experimental situations described (i.e. in presence of pH modifying yeast, etc.; to which degree is the 'trap and transfer' method (cf. line 440) reliable and also accounting for 'hard to trap' and poorly abundant bacteria?).

In the intention to cover a broad range of BFI and their ecology (cf. the generalizing title of the manuscript), the manuscript presents a wealth of relevant data while simultaneously asking too many (unanswered) questions. It thereby remains in parts imprecise (missing controls, missing discrimination between growth and dispersal, missing definitions (what is bacterial motility?),

etc.), mostly descriptive and may remain beyond generic insights. The work hence may be better suited for a more specialized scientific journal.

Thank you for the opportunity to submit a significantly revised version of our manuscript “**Fungal networks shape dynamics of bacterial dispersal and community assembly in cheese rind microbiomes**” The reviewers provided excellent feedback on our work and we appreciate their insightful comments. We believe that our revised manuscript is much stronger after incorporating their suggestions.

Three major issues emerged from the reviewers' remarks:

- 1) There were sections in our manuscript that lacked clarity or were missing information. These sections caused confusion for the reader. As we have noted in a detailed response to reviews below, we have corrected all of these issues by adding more details, providing new data, and re-writing sections of our paper.
- 2) Both reviewers suggested that some of our claims about the potential roles of fungal networks were too general. We have reshaped the scope of our work by changing the title of our manuscript and by rewriting parts of the Abstract and Discussion.
- 3) Both reviewers thought that our community experiments needed a better control to determine the relative impact of physical networks (abiotic) versus the activity of the fungus (biotic) through other mechanisms (changing the pH, releasing free amino acids, etc.). We have conducted new experiments that demonstrate that the presence of physical networks alone is enough to cause shifts in community composition.

We address these issues and other more minor points in a detailed response below. The original reviewers comments are in bolded text.

Reviewer #1 (Remarks to the Author):

The MS by Zhang et al investigates the effect of three different fungal networks first on the dispersal of *Serratia proteamaculans*, and afterwards in the dispersal of an assemblage of a few bacterial strains including the same strain indicated previously.

The manuscript presents results from different types of experiments, including the comparison of dispersal on biotic/abiotic networks, mutagenesis and transcriptomics, and manipulation of (simplified) bacterial assemblages. Although this is highly commendable, the message of the manuscript is unfortunately not clear. In addition, the conclusion in terms of the demonstration of a community-level effect of fungal dispersal networks is (in the opinion of this reviewer) an overstatement. One fungal species and a handful of bacterial strains do not (by far) make a microbial community (although I acknowledge that this is more complex than two-species interactions). This point is one of the main aspects put to the fore in the introduction to the manuscript, but I consider that the authors should be more critical.

We agree with the reviewer that our cheese rind microbial communities are relatively simple and low in diversity. However, we disagree with the statement “one fungal species and a handful of bacterial strains do not (by far) make a microbial community.” An ecological community is

defined as a group of two or more species growing together in a particular habitat. Groups of microbial species do not need to be remarkably complex and they do not need to grow in undisturbed or natural environments to be considered a community.

We have shown in previous work that many cheese rinds harbor just a few (3-10) bacterial and fungal species and reproducible cheese rind communities (with similar species compositions) form in very different geographic locations (Wolfe et al. 2014). Many of these communities are dominated by just a single filamentous fungus, making our single fungal network experiments relevant to real-world cheese communities. While many soil and human gut microbiomes have much more complex microbial communities, other systems like some insect guts, skin surfaces of mammals, and biofilms in built environments have similar levels of microbial diversity. Therefore, we believe that our cheese rind microbiomes are useful models for low complexity microbial communities.

We agree with the reviewer that the overall framing of the first submission may have overstated the effects of fungal networks on microbial community composition and tried to generalize these findings to more complex communities. To address this concern, we have reshaped the messaging of the paper to be more specific to cheese rind microbiomes and more carefully discuss how our results could extend to other systems in the Discussion. We have changed the title of the manuscript from “Fungal networks mediate microbiome composition through dispersal facilitation of motile bacteria” to “Fungal networks shape dynamics of bacterial dispersal and community assembly in cheese rind microbiomes.” We are also more clear in the discussion section that the impact of fungal-mediated bacterial dispersal that we describe in our paper may not be the same in more complex communities.

We have also added another dataset to our paper to address the reviewer’s concern of limited species diversity (Figure 4a). We have screened the dispersal of all dominant bacterial species from cheese rinds (23 species/strains) across the three fungal networks (*Mucor*, *Galactomyces*, and *Penicillium*). The bacteria screened span a wide range of bacterial taxa and lifestyles and can be found in non-cheese environments (e.g. mammalian skin). These data demonstrate that almost all motile Proteobacteria use fungal networks for dispersal. As far as we are aware, this is the largest screen of bacterial dispersal on fungal networks conducted to date.

In addition, it is difficult to form a clear idea of the hydration level of the different media used (with a few exceptions). Additional information is required here, as this abiotic factor is very important when evaluating the relative contribution of fungus-facilitated dispersal versus colony expansion (which appears in the case of the strain used to play a very significant role on dispersal).

In the case of Figure 1, it is unclear which medium was used for the observation of the “unusual streaming colonies” (parts b and C). The same applies for figure d. In this case, indicating the hydration level of the medium is very important. In addition, if information concerning the matric potential of the cheese rind is available, this should be included.

We appreciate the reviewer pointing out the importance of water availability in driving these interactions, and we strongly agree with this point. To better clarify the hydration level of the various media used, we report water availability for cheese, cheese curd agar, brain heart infusion agar (BHI agar) and plate count agar with milk and salt (PCAMS) in the methods section of our revised manuscript (see end of first paragraph of “Network Dispersal Co-Spot Assays” section). These are the main types of substrates used throughout the paper. Water activity is the standard metric that food microbiologists use to describe hydration levels in food matrices, so we decided to use this as our metric of hydration level. We also more clearly state which type of medium is used in the various figures or assays by indicating this in the figure legend or Methods section.

Supplementary Figure 1 is not related to the information given in the text. The figure presents the phylogeny of the strains isolated, not the formation of bacterial streams. Please refer to this supplementary figure in the material and methods.

We apologize for the confusion here. We cited it early in the paper because the phylogeny helps clarify the species of bacteria and fungi used in the experiments. But we agree that it would make more sense to refer to this in the methods section. This is now Supplementary Figure 10 and is noted in Methods.

For figure 2 one has to understand from the description of the methods that the fungi were inoculated as spores (is this right?) and simultaneously with the bacterial cells (which supposes the use of a liquid substrate to place both in the center of the Petri dish and thus a boost on bacterial dispersal by colony expansion; see for example comparison of figure 1a with 1c, which precisely correspond to a pre-form dispersal network). In this experiment one has to control for the effect of the bacterial on fungal germination and the density of the network between the different species. This is partially considered for what the authors call a static network. However, in the images the extent of the fungal network in each of the treatments is not indicated (using the same timing is not necessarily relevant and comparable in terms of the biology of the fungal species used). This needs to be indicated.

We apologize for the confusion here. The reviewer is correct. Fungi were inoculated as spores with the bacteria in liquid PBS. However, the spot of liquid dried out within <10 minutes after inoculation, so it did not impact initial dispersal dynamics. Both fungal spores and bacterial cells originated from frozen glycerol stocks, so both bacterial and fungal cells were in dormant states when inoculated onto the plate. We have clarified these details in the methods section.

In the static network experiments, the fungal network was distributed across the entire plate at an equal density of colony forming units and inoculated at the same time. By the 48 hour time point, all three fungi had formed lawns of mycelia at equal densities across the entire surface of the plate. We have indicated this in the figure legend.

We agree the fungi we used in our experiments have have different mycelium densities. But we don't think that this necessarily confounds these experiments. The density of the fungal network is what is driving the differences in bacterial dispersal across the networks.

One aspect that surprises me is the effect of *Serratia* on sporulation. I am quite surprised at observing no spores in either the *Mucor* or the *Penicillium* systems. Could the authors indicate their observations (for example see supplementary Figure 4)? This is particularly relevant when the authors discuss the effect on fitness based on CFU counting, and the apparent inhibitory effect discussed in the text. Please consider including values of total active biomass as a parallel measure to fitness.

We appreciate this comment from the reviewer and understand their sense of surprise. In Supplemental Figure 4 (which is now Supplemental Figure 3), we only show *Mucor* because we focused only on *Mucor* in this section of the paper. In this figure, *Mucor* is producing some spores, but they are not darkly pigmented in this photo because it is growing on brain heart infusion (BHI) agar. We used BHI agar for all of our interaction screens because it allows for both the bacteria and the fungi to grow well. BHI represents optimal conditions for these interactions. *Mucor* and other fungi do make more obvious spores when growing on other substrates, including the PCAMS used in our Tn5 transposon screen (see Supplemental Figure 6). In other experiments (for example those presented in Figure 1d), spores were not produced by any fungi because it was relatively early in their growth cycles.

We did not measure total active biomass (such as ergosterol). We use CFUs as a measure of fitness because it captures total number of reproductive structures (spores + hyphal fragments) and not just total biomass and it is a measure we can use across bacteria and fungi (both filamentous fungi and yeasts).

The experiment attempting to identify mechanistic elements of the interaction through mutagenesis is particularly interesting, however, the results presented are disappointing. The authors focus on only one of the mutants. The reader wonders what are the genetic modifications of the other mutants with a modified phenotype. In addition, when referring to the role of flagella in dispersal, please use all the existing references in the topics (this is something already established by other groups in the past).

We appreciate the enthusiasm for our mutagenesis screen and interest in the other mutants that we obtained from our screen. We focused on the mutant with strongly impaired motility (Tn5_13) because it was most relevant to the storyline of the paper (dispersal on fungal networks). We agree that the other mutants, including those that completely inhibited fungal growth, are fascinating. We have included a more detailed discussion of these mutants in the text (see the sixth paragraph in the section "RNA-seq, transposon mutagenesis, and

comparative genomics...”) and provide more info on the phenotypes of these mutants in Supplemental Figure 6.

We agree that other groups have indicated that flagella can play a role in the dispersal of bacteria on fungal hyphae. We have added appropriate references to these papers in our section discussing the Tn5 transposon results. It is important to note that these previous studies took a very directed approach and only considered flagella mutants. They did not do unbiased screens and just assumed flagella biosynthesis is one of the most important mechanisms driving bacterial dispersal on fungal networks. Our transposon mutagenesis screen is the first to use an unbiased screen to identify key regulators of this interaction.

The experimental set-up of the experiments with artificial communities is unclear. Were the bacterial strains co-inoculated in a static or in an active network? In addition, it is unclear how the relative abundance of each bacterial strain was measured. Was this based on differences in colony morphology? Please clarify all these elements. In addition, please add a control for the artificial communities without dispersal network in order to assess the effect of competition regardless of dispersal (your strains are likely to all have different growth rates, which could alter the ratios obtained regardless of dispersal). The bar corresponding to input is not defined.

We agree that more detail on how these experiments were set up would provide improved clarity in our manuscript. We have revised the Methods section of the paper to provide this clarity. The bacterial communities were co-inoculated with fungi that then grew together with the bacteria. This would be similar to the active networks.

Relative abundance of the different bacterial species was determined by harvesting the entire bacterial community and then plating on selective media. All bacterial species have unique colony types and can be easily differentiated by colony morphology. We have clarified this in the methods and have included a photo of colonies of the four bacterial species (Supplemental Figure 11).

In terms of the control community, we believe that our “No Network” treatment is this control. The No Network treatment was just the community of bacteria and the yeast without the addition of filamentous fungi. “Input” was the relative abundance of the different bacterial species that were initially added to the community. We have clarified all of these points in the Methods and Figure legend, and we are grateful to the reviewer for pointing out the lack of clarity here.

Finally, the authors should revise all figure legends (including the supplementary) for elements missing in the description. These include: media used, undefined terms (e.g. N.D. SFig 5; -GW/+GW SFig 7), and scales for the images.

We appreciate the feedback on our figure legends and figures and have made all of the suggested revisions.

Reviewer #2 (Remarks to the Author):

Zhang et al. analyze the effect of fungal mycelia on the dispersal, gene expression and community composition of selected bacteria isolated from cheese rinds. In synthetic microcosms they reveal that mycelia allow for and differentiate bacterial dispersal and growth of bacteria and thereby enable temporal shifts of an (artificial) bacterial community. By performing RNA-seq., transposon mutagenesis and comparative genomic analysis they find that the presence of *Mucor lanceolatus* may not influence global gene expression of flagellar biosynthesis (or other genes/regulators) yet rather lead to upregulation of genes related to metabolic processes.

The authors thereby confirm many previously known concepts/facets of bacterial-fungal interactions (BFI) and show that many BFI also may take place in the microbial habitat of cheese rinds: e.g. i) the role of mycelia (and mycelial properties on bacterial dispersal and growth of both synthetic and more complex microbial communities, ii) the influence of bacterial motility on dispersal, iii) the presence of metabolic interactions of bacteria and fungi in the hyphosphere, iv) analyses of complex bacterial fungal communities (including molecular and genetic analyses), etc. Many of the claims of novelty of the manuscript hence may be overstatements.

We acknowledge that some aspects of our work have been partially demonstrated in other systems. But much of our work remains novel, including the cellular-scale imaging, the transposon mutagenesis screen, and the community-level experiments. We show the relative importance of this bacterial-fungal interaction (bacterial motility on fungal hyphae) in a specific and integrated ecological context. All previous studies on this type of interaction have used strains of bacteria and fungi that do not necessarily co-occur in nature. Therefore, the ecological significance and consequences of this previous work is unclear. Our cheese rind system is a widespread microbial community that can be easily manipulated and with a defined ecological context.

To address this reviewer's point, we have toned down writing throughout the paper as we indicated above.

Interesting and novel insights give the study's RNA-seq, transposon mutagenesis and comparative genomic analysis. These point at a downregulation of the chitinase genes and possible metabolic interactions of the *Serratia* with the fungus, rather than at an upregulation of dispersal-related genes. Although following a promising approach the given experiments seem to open more questions than answers; relevant additional experiments are missing to give transferable insights to the BFI ecology: e.g. to which degree are the changes due to altered nutrient status of the bacteria (i.e. independent of the presence of a fungus)? Do similar changes also occur in presence of other (dispersal

promoting; not dispersal promoting) mycelia? To which degree are such up-/down-regulation dependent on the age of the fungal mycelia (cf. data in Fig. 2f)?

We agree with the reviewer that as with any scientific study, our RNA-seq data generate new questions. But they also provide important answers about how genes are differentially expressed during this interaction. Specifically, they clearly show no changes in the expression of genes associated with motility when bacteria disperse on fungal networks, while metabolism genes are differentially expressed. We did not attempt to, nor claim to, mechanistically dissect any results from the RNA-seq dataset. Instead, we integrated a suite of tools to provide a broad systems-level perspective of an interesting interaction in an ecologically and culturally significant microbiome, and highlights the potential ecological consequences of this interaction. We believe that the breadth that comes from the integration of tools and datasets in our paper is more important than the depth that would be obtained from drilling into one specific aspect of the paper. Future studies and papers will explore in more detail the findings from this systems-level view, including following up on these RNA-seq observations.

Similar resolution problems may also be observed in the experiments using synthetic multispecies microbiomes. While following an interesting approach, experimental data unfortunately does not allow discriminating the influence of dispersal from the influence of growth (only relative data of the biomass and no spatio-temporal resolution of the microbiome are provided; a control with artificial networks allowing for spatiotemporal distribution in the absence of bacterial fungal interactions (BFI) is missing). The data rather remain descriptive lacking in depth ecological insights beyond the experimental situations described (i.e. in presence of pH modifying yeast, etc.; to which degree is the ‘trap and transfer’ method (cf. line 440) reliable and also accounting for ‘hard to trap’ and poorly abundant bacteria?).

We appreciate this reviewer's concern about appropriate controls in the synthetic microbiome experiments. We agree that the addition of a non-biological network treatment could help to tease apart the abiotic and biotic contributions. We have included data from a new experiment that we conducted to address this reviewer's concerns (Figure 4d). We repeated the same community experiments, but instead of adding living fungal networks, we added synthetic networks (glass fibers). We predicted that relative abundance of *Serratia* within the bacterial community would increase in response to the synthetic networks, just as it does to the living *Mucor* networks. This is exactly what we found. There are some caveats to using glass wool to perfectly mimic the presence of fungal networks, and we clearly point them out in our revisions.

In terms of how well our method measures the relative abundance of the different bacteria, we have clarified how we measured relative abundances in the methods. We did not use the tap and transfer approach for these experimental communities, but instead harvested the entire community, homogenized it, and the plated it out to obtain CFUs of the community members. The different bacterial species are clearly distinguished based on their colony morphologies as we now more clearly indicate in this revised manuscript (see Supplementary Figure 11). This is

a robust approach to measure living bacterial cells and measure community composition, and is something we have used in previous publications (Wolfe et al. 2014, *Cell*; Kastman et. al. 2016, *mBio*).

In the intention to cover a broad range of BFI and their ecology (cf. the generalizing title of the manuscript), the manuscript presents a wealth of relevant data while simultaneously asking too many (unanswered) questions. It thereby remains in parts imprecise (missing controls, missing discrimination between growth and dispersal, missing definitions (what is bacterial motility?), etc.), mostly descriptive and may remain beyond generic insights. The work hence may be better suited for a more specialized scientific journal.

We appreciate this reviewer's overall impressions of our work, but wish they provided more detail in their critique so we could adequately respond to their comments. We would love to know more about what they mean by 'missing discrimination between growth and dispersal' and 'missing definitions (what is bacterial motility?), etc.'. These points are mentioned in passing in their note and the issues they have are not fully explained.

Our definition of bacterial motility was very clear in our first submission.

"Small-scale (micrometer-centimeter) dispersal of bacterial cells is one key ecological process that may determine the dynamics of microbiome assembly. After a propagule (cell, spore, etc.) of a microbial species colonizes a potential habitat, the ability to grow and rapidly colonize available niche space may determine both the distribution and functions of that particular species within the community. Many bacteria are known to use active motility, via extracellular appendages or secreted metabolites, to disperse over small spatial scales up or down gradients of resources or attractants. A significant body of work from just a few model bacterial species has determined the genetic and biophysical mechanisms of active bacterial dispersal, including swimming, swarming, gliding, twitching, and sliding."

As noted above, we have toned down some of the language in our paper (including changing the title) to indicate that our findings are specific to cheese rind microbiomes and that it may be challenging to generalize all of our results to other systems. We have also tried to highlight the integration across scales in our manuscript to focus on the idea that breadth is a strength of our work, not a weakness. We've also included a conceptual summary figure to further highlight this integration (Figure 5).

To address the comment about the manuscript reaching too far with too much data, we have trimmed some experiments from the first descriptive section of the paper that were not important to the main theme of the paper. Specifically, we removed the section of the stability of the interaction between *Serratia* and *Mucor*. We also worked hard to streamline the text (especially in the Introduction and first Results section) to highlight the integration of our work and to improve clarity for the reader.

Reviewer #1 (Remarks to the Author):

Overall the authors made a very thorough revision and provided enough information in their rebuttal. After the revision the manuscript is much easier to read and to understand. I would suggest however an editorial revision of some minor aspects such as:

1. Use of the word "Discovery" in the legend of Fig. 1. This word does not add info to the legend.
2. Better clarifying the choice of CFU measurements as an indication of fitness. The explanation in the rebuttal would be useful for general readers.

Reviewer #2 (Remarks to the Author):

The revisions (in particular better focus on the ecology of bacteria-fungus interactions (BFI) in cheese rinds) have improved the manuscript by Zhang et al. Many former generalizing 'overstatements' have been removed and necessary additional experiments have been added. The added value of the new Figure 5, however, seems limited.

The manuscript may still profit from a more careful discussion of the findings in terms of previous work on BFI. For instance, work e.g. by Warmink et al.; Nazir et al., (2012), Simon et al. (2017) have shown using molecular biological tools effects of hyphae the shaping of microbial communities in the hyphosphere (cf. your statement on lines 61ff or 287ff does not reflect current knowledge; In this context it may be useful to tone down/omit lines 7-10 in the abstract). Likewise streamlining statements on lacking knowledge on biophysical drivers shaping bacterial dispersal along hyphae (line 158f; please c.f. e.g. Taissa Vila et al, 2016) or the effect of small scale dispersal on microbial evolution and diversity (lines 365 ff; cf. e.g. Zhang et al. 2014; Berthold et al., 2016, de Carvalho et al. 2015; lines 372ff; cf. Nazir 2009) may be avoided or discussed more carefully.

Line 17f: Why? Please explain why dense and slow-growing mycelial networks may not allow for efficient bacterial dispersal? Which is the correlation between fungal growth rate, network density and dispersal?

Line 147: Why is the sum of spores and hyphae fragments (units that form cfu in your test) a fitness parameter for fungi. Is spore formation a fitness indicator? Why did you not quantify the fungal biomass?

Lines 299f: please mention that these bacteria were isolates from cheese rinds.

Line 375: What do you mean by 'unbiased screens'. This statement hides, that hypotheses and data on this or similar research questions already exist.

Line 393: here, it may be interesting to discuss an earlier finding that fungal hyphae indeed promote food spoilage (cf. Ken-ichi Lee, 2014).

Reviewer #1 (Remarks to the Author):

Overall the authors made a very thorough revision and provided enough information in their rebuttal. After the revision the manuscript is much easier to read and to understand. I would suggest however an editorial revision of some minor aspects such as:

1. Use of the word "Discovery" in the legend of Fig. 1. This word does not add info to the legend.

We have made this suggested edit.

2. Better clarifying the choice of CFU measurements as an indication of fitness. The explanation in the rebuttal would be useful for general readers.

Both reviewers requested clarity in the manuscript about using CFUs as an indicator of fitness. Fitness is a loaded term that means many different things to many different sectors of biology. We do believe that the total number of colony forming units (CFUs) is an excellent measure of reproductive potential and works well for both filamentous fungi and bacteria. It is a standard practice in the field to use CFUs for both fungi and bacteria. But to avoid confusion in the manuscript, we have removed all references to fitness and have replaced them with "growth as indicated by the number of colony forming units (CFUs)" or "growth." We have provided a brief justification of this approach in the methods (see "Co-culture growth assays" section).

Reviewer #2 (Remarks to the Author):

The revisions (in particular better focus on the ecology of bacteria-fungus interactions (BFI) in cheese rinds) have improved the manuscript by Zhang et al. Many former generalizing 'overstatements' have been removed and necessary additional experiments have been added. The added value of the new Figure 5, however, seems limited.

We are glad that the reviewer appreciates our manuscript edits. We find that summary figures such as Figure 5 are useful in manuscripts like ours that have diverse datasets and messages. We like to leave the reader with a clear visual summary of the work we did and the main take-home messages. These summary figures can also be useful for general audiences not familiar with features of our systems. So we would like to leave the figure in the manuscript unless the editor thinks it would be best to remove.

The manuscript may still profit from a more careful discussion of the findings in terms of previous work on BFI. For instance, work e.g. by Warmink et al.; Nazir et al., (2012), Simon et al. (2017) have shown using molecular biological tools effects of hyphae the shaping of microbial communities in the hyphosphere (cf. your statement on lines 61ff or 287ff does not reflect current knowledge; In this context it may be useful to tone

down/omit lines 7-10 in the abstract). Likewise streamlining statements on lacking knowledge on biophysical drivers shaping bacterial dispersal along hyphae (line 158f; please c.f. e.g. Taissa Vila et al, 2016) or the effect of small scale dispersal on microbial evolution and diversity (lines 365 ff; cf. e.g. Zhang et al. 2014; Berthold et al., 2016, de Carvalho et al. 2015; lines 372ff; cf. Nazir 2009) may be avoided or discussed more carefully.

We appreciate this reviewer's enthusiasm for citing even more papers in our manuscript. However, we are already over the limit of citations we can include. We are aware of the excellent studies they note and we cite many key papers on bacterial-fungal interactions in both the introduction, results, and discussion.

We believe that our statements on lines 61 and 287 do indeed accurately reflect the current understanding of bacterial dispersal on fungal networks. As we state in line 61 "Previous studies characterizing fungal-mediated bacterial dispersal relied on artificial combinations of bacteria and fungi with unknown natural histories and limited ecological context." and in line 287 "Previous studies have described the potential existence of fungal-mediated bacterial dispersal in soil systems through pairwise interaction studies of a few laboratory strains." Our goal with these statements it to note that most previous studies of bacterial dispersal on fungal hyphae had used strains that were not co-isolated (isolated form the same sample/environment/habitat) and may not necessarily interact in nature. We are not downplaying or denigrating this previous work, but are trying to distinguish our work which starts from the observation of a bacterium growing on fungal hyphae.

Our statement in line 158 is "But the initial phases of colonization occur at the micron scale and the relevant biophysical interactions that regulate bacterial dispersal on hyphae at these scales are **unknown**." We have changed it to "But the initial phases of colonization occur at the micron scale and the relevant biophysical interactions that regulate bacterial dispersal on hyphae at these scales are **poorly characterized**."

Our statement in lines 7-10 of the abstract is "Most studies of bacterial motility have examined rates and modes of small-scale (micrometer-centimeter) cell dispersal in monocultures." We strongly stand by this statement. If you summed up all of the work on bacterial motility, a vast majority of it is done with just *E. coli*, *Pseudomonas*, and other model bacterial strains in a Petri dish. Again, we do acknowledge that there are studies like ours that have explored how interactions can shape dynamics of bacterial dispersal and we cite these throughout the paper.

Line 17f: Why? Please explain why dense and slow-growing mycelial networks may not allow for efficient bacterial dispersal? Which is the correlation between fungal growth rate, network density and dispersal?

It was not the goal of our manuscript to explain why bacteria disperse at different rates on different fungal networks and we are not comfortable speculating as to why this might be. We have removed this statement from the abstract.

Line 147: Why is the sum of spores and hyphae fragments (units that form cfu in your test) a fitness parameter for fungi. Is spore formation a fitness indicator? Why did you not quantify the fungal biomass?

Measuring CFUs to quantify amounts of bacteria and fungi is standard practice in the field. Fungal biomass is not necessarily a good indicator of total reproductive potential or fitness because some components of biomass won't necessarily contribute to reproduction and those that do contribute disproportionately to reproduction (e.g. conidiophores) may not have as much biomass as other structures. As we noted above, we have changed our language in the section discussing fitness to avoid these issues. We have also provided a rationale statement in the methods for our approach.

Lines 299f: please mention that these bacteria were isolates from cheese rinds.

We appreciate the reviewer pointing out this lack of clarity. We have added text to this sentence to clarify that these are cheese rind isolates.

Line 375: What do you mean by 'unbiased screens'. This statement hides, that hypotheses and data on this or similar research questions already exist.

We apologize for the lack of clarity with this statement and misinterpretation. And we definitely do not hide that hypotheses and data on similar research already exists as we cite numerous papers in the same sentence that present those hypotheses and data. We also state earlier in the paper "our screen supports previous targeted knockout studies and confirms a key role of flagella in fungal-mediated bacterial dispersal^{26,45,55}." To make sure other readers don't misinterpret our statements, we have reworded this section to attempt to be more clear. We have changed this section from:

"Several previous studies have demonstrated that motile bacteria can migrate on fungal hyphae^{24,26,28,29,45}, but the genetic mechanisms of these interactions have not been identified using unbiased screens. Our novel interaction-based transposon mutagenesis screen determined that flagella-mediated motility is essential for the dispersal of *Serratia proteamaculans* in liquid layers on fungal hyphae (Fig. 5b)."

to

"Several previous studies have demonstrated that motile bacteria can migrate on fungal hyphae^{24,26,28,29,45}, but the genetic mechanisms of these interactions are **not well-characterized and are often examined based on the selection of a**

few candidate genes. Our novel interaction-based transposon mutagenesis screen **used an untargeted approach to identify any non-essential genes controlling this interaction.**” We determined that flagella-mediated motility is essential for the dispersal of *Serratia proteamaculans* in liquid layers on fungal hyphae (Fig. 5b).

Line 393: here, it may be interesting to discuss an earlier finding that fungal hyphae indeed promote food spoilage (cf. Ken-ichi Lee, 2014).

This is an interesting idea, but once again the number of references we can include limits us.